# Functional and Stress Response Analysis of Heat Shock Proteins 40 and 90 of Giant River Prawn (*Macrobrachium rosenbergii*) under Temperature and Pathogenic Bacterial Exposure Stimuli

**DOI:** 10.3390/biom11071034

**Published:** 2021-07-15

**Authors:** Tanya Ju-Ngam, Nichanun McMillan, Mamoru Yoshimizu, Hisae Kasai, Ratree Wongpanya, Prapansak Srisapoome

**Affiliations:** 1Laboratory of Aquatic Animal Health Management, Department of Aquaculture, Faculty of Fisheries, Kasetsart University, Chatuchak, Bangkok 10900, Thailand; radin_the@hotmail.com; 2Center of Advanced Studies for Agriculture and Food, Kasetsart University Institute for Advanced Studies, Kasetsart University, Bangkok 10900, Thailand; 3Center of Excellence in Aquatic Animal Health Management, Faculty of Fisheries, Kasetsart University, Chatuchak, Bangkok 10900, Thailand; 4Laboratory of Aquaculture Genetics, Department of Aquaculture, Faculty of Fisheries, Kasetsart University, Chatuchak, Bangkok 10900, Thailand; ffisnnp@ku.ac.th; 5Laboratory of Marine Biotechnology and Microbiology, Faculty of Fisheries Sciences, Hokkaido University, Hakodate 041-8611, Japan; yosimizu@fish.hokudai.ac.jp (M.Y.); hisae@fish.hokudai.ac.jp (H.K.); 6Department of Biochemistry, Faculty of Science, Kasetsart University, Bangkok 10900, Thailand; fscirtw@ku.ac.th

**Keywords:** *Macrobrachium rosenbergii*, heat shock protein, Hsp40, Hsp90, *Aeromonas hydrophila*, heat–cold temperature shock, gene silencing

## Abstract

The aims of this research were to perform molecular characterization and biofunctional analyses of giant river prawn *Hsp40* and *Hsp90* genes (*Mr-hsp40* and *Mr-hsp90*) under various stress conditions. Comparisons of the nucleotide and amino acid sequences of *Mr-hsp40* and *Mr-hsp90* with those of other species showed the highest similarity scores with crustaceans. Under normal conditions, expression analysis using quantitative real-time RT-PCR (qRT-PCR) indicated that *Mr-hsp40* was highly expressed in the gills and testis, and *Mr-hsp90* expression was observed in all tissues, with the highest expression in the ovary. The expression patterns of *Mr-hsp40* and *Mr-hsp90* transcripts under *Aeromonas hydrophila* challenge and heat–cold shock conditions were examined in gills, the hepatopancreas and hemocytes, at 0, 3, 6, 12, 24, 48 and 96 h by qRT-PCR. Under bacterial challenge, *Mr-hsp40* displayed variable expression patterns in all tissues examined during the tested periods. In contrast, upregulated expression of *Mr-hsp90* was quickly observed from 3 to 12 h in the gills and hepatopancreas, whereas obviously significant upregulation of *Mr-hsp90* was observed in hemocytes at 12–96 h. Under temperature shock conditions, upregulation of *Mr-hsp40* expression was detected in all tested tissues, while *Mr-hsp90* expression was quickly upregulated at 3–48 h in all tissues in response to 35 °C conditions, and conditions of 35 and 25 °C stimulated its expression in gills and the hepatopancreas at 12 and 48 h, respectively. Silencing analyses of these two genes were successfully conducted under normal, high-temperature (35 °C) and *A. hydrophila* infection conditions. Overall, knockdown of *Mr-hsp40* and *Mr-hsp90* effectively induced more rapid and higher mortality than in the PBS control and GFP induction groups in temperature and infectious treatments. Evidence from this study clearly demonstrated the significant functional roles of *Mr-hsp40* and *Mr-hsp90*, which are crucially involved in cellular stress responses to both temperature and pathogenic bacterial stimuli.

## 1. Introduction

*Macrobrachium rosenbergii,* or giant river prawn, is an economically important crustacean of central and southeastern Asia. Hatching and farming of this prawn is rapid, and culture has been developed in many countries due to its good flesh, delicate texture and palatable taste [1]. Moreover, this prawn has survived via adaptation to broad temperature changes from approximately 14 to 35 °C [2]. In contrast, similar to other aquatic animals, *M. rosenbergii* production has been decreased by various problems, especially rapid changes in culture conditions that usually induce prawns to become stressed and easily infected with many pathogens [3,4,5]. These stresses have negative effects on their bodies, and the organisms respond by releasing many hormones and proteins to prevent critical damage to cellular homeostasis [6]. One of the important defense mechanisms against various stressors is known to rely on the release of heat shock proteins (Hsps) [7].

Hsps are a group of innate immune proteins essential for managing stresses, and their functions as molecular chaperones include repairing all denatured proteins to their normal form by protein refolding [8]. Hsps are ubiquitous, highly conserved and found in all living cells, from bacteria to higher vertebrates, and they can be stimulated by many stressors, such as pathogen infections, physical system disruption and extreme environmental impacts [9,10]. Activities of cellular life, such as cell processes, growth and development and apoptosis, result in reduced levels of Hsp expression [11]. Basically, Hsps are generally classified by their structure, molecular weight and functions, and they include Hsp100, Hsp90, Hsp70, Hsp60, Hsp40 and several smaller Hsps [12].

Hsp40s, or chaperone DnaJs, have an evolutionarily conserved structure and are cochaperones with Hsp70 proteins, giving rise to the ATPase activity of Hsp70, which play important roles in controlling protein synthesis, folding, processing and degradation [13]. Generally, all Hsp40 family members consist of approximately 70 amino acid residues in the J domain that specifically bind to the Hsp70 ATPase domain [14]. The J domain is a significant part of Hsp40s that interacts with their partners to enhance correct protein folding and refolding [15]. Additionally, Hsp40s can be divided into three clusters: Hsp40 type I, which has a glycine/phenylalanine-rich (G/F) domain, cysteine-rich Zn^2+^ binding (CR) domain and C-terminal domain (CTD), Hsp40 type II, which has no G/F domain, and Hsp40 type III, which has only a CTD [16].

Hsp90s are a massive group of molecular chaperones that have been mainly found in cellular stress responses and represent 1–2% of total cytosolic proteins under normal conditions but can be increased up to 4–6% by stress stimuli [17]. Hsp90 functions as a client protein to induce the activation of signal transduction proteins, such as steroid hormone receptors, cell cycle kinase Cdk4 and serine/threonine kinases [18,19]. Hsp90s are known to play pivotal roles in protein folding and degradation, preventing protein misfolding and aggregation, and they transport proteins to targeted organelles [20,21]. In addition, Hsp90 can inhibit the transcription of other Hsps by controlling the heat shock protein factor 1 (HSF1) [22]. The Hsp90 protein structure is composed of three domains: a N-terminal domain, a core or middle domain and a CTD [23]. Hsp90 alone is unable to activate its functions and naturally forms a dimer with other Hsp90s or chaperone molecules using a C-terminal tetratricopeptide repeat (TPR)-recognition motif [24]. A previous study supported the hypothesis that Hsp90 stabilizes denatured proteins and works together at the ATP-dependent site of the Hsp70-Hsp40 chaperone system to refold those proteins [25].

At present, Hsps have been increasingly investigated in aquatic animals, especially decapods, because their functions directly affect the innate immune response in all living organisms. Moreover, rapid alteration of environmental or culture conditions causes fundamental negative effects, inducing easier infection with many pathogens, and Hsps are thus stimulated to help protect host cells against those stressors. Additionally, during cellular stress responses, aberrant secreted proteins can be destroyed by ER-associated protein degradation (ERAD), and a prominent, medically relevant ERAD substrate is the cystic fibrosis transmembrane conductance regulator (CFTR). In this process, Hsp90 crucially involves maintaining CFTR structural integrity, whereas Hsp40 catalyzes CFTR degradation [26]. Therefore, to better define the chaperone requirements and understand crucial functional roles of Hsp40 and Hsp90 of the giant river prawn (*M. rosenbergii*), cDNAs encoding the *Hsp40* and *Hsp90* genes were cloned and characterized in the present study. Their expression was evaluated after pathogenic bacterial challenge and heat–cold stress exposure. Additionally, a gene-silencing technique that is normally used as a crucial research tool to study various target gene functions was employed to elicit their critical roles under both high-temperature and bacterial infection conditions.

## 2. Materials and Methods

### 2.1. Experimental Animals

In this study, two hundred healthy giant river prawns (40–50 g) were obtained from a private farm in Nakornprathom Province, central Thailand. The prawns were acclimatized in a 1000 L fiberglass tank containing a clean fresh aeration system and maintained at 30 ± 2 °C for 7 days. During this period, the prawns were fed three times per day with commercial feed at 3–5% body weight.

### 2.2. Total RNA Isolation and mRNA Extraction

To isolate total RNA and mRNA, 1.5 mL of prawn hemolymph was withdrawn from the ventral sinus using a 3 mL syringe with a 24G needle containing 1.5 mL of anticoagulant solution (10% trisodium citrate in RPMI-1640 medium, Sigma-Aldrich, St. Louis, MO, USA). The hemolymph and anticoagulant mixture was transferred into an Eppendorf tube and centrifuged at 7500 rpm for 15 min at 25 °C. Hemocyte pellets were further harvested for the next step. After that, the prawns were dissected, and the eyestalk, foregut, gills, heart, hepatopancreas, hindgut, midgut, muscle, subcuticular epithelium, testis, thoracic ganglion and vas deferens from a male and ovary from a female prawn were collected. All samples were preserved in TRIzol reagent (Gibco, Waltham, MA, USA) and used for RNA extraction according to the manufacturer’s instructions. The RNA pellet from all tissues was air dried, and total RNA was dissolved in sterile distilled water. The total RNA concentration and quality were examined at 260 and 280 nm by an iMark™ Microplate Absorbance Reader (Bio-Rad, Hercules, CA, USA). The total RNA samples were immediately treated with DNase I (Fermentas, Waltham, MA, USA) to eliminate genomic DNA contamination. The mRNA from hemocytes, the hepatopancreas and muscle were consequently prepared with a QuickPrep Micro mRNA Purification Kit (Amersham Biosciences, Piscataway, NJ, USA) according to the manufacturer’s instructions.

### 2.3. 5′ Rapid Amplification of cDNA Ends (RACE) PCR

Sequence analysis of two partial cDNAs retrieved from a hemocyte EST library of the giant river prawn clones P2714 (EL696523) and BG21346 (EL609971) showed a high similarity to the hsp40 and hsp90 genes respectively, of other animal species available in the GenBank database. Since they contain only information at the 3′ downstream end of the full sequences, the specific primers R_GFP_*hsp40* and R_GFP_*hsp90* were designed to recover full-length cDNAs of these cDNA clones (Table 1). The 5′ RACE first-strand cDNA was synthesized using mRNA isolated from giant river prawn hemocytes as templates. One microgram of the mRNA was converted into 5′ ready-to-use RACE first-strand cDNAs using a BD Smart^TM^ RACE cDNA Amplification Kit (BD Biosciences, Clontech, Palo Atlo, CA, USA). This first strand was further used as a template to amplify the 5′ end of the target genes. Each RACE PCR for each primer pair (final volume of 25 µL) consisted of 5′ RACE PCR conditions, and touchdown PCR steps strictly followed the manufacturer’s instructions. Briefly, the first five cycles were run at 94 °C 30 s and 72 °C 3 min. The second 5 cycles were amplified at 94 °C 30 s, 70 °C 30 s and 72 °C 3 min. Finally, the other 27 cycles of 94 °C 30 s, 68 °C 30 s and 72 °C 3 min were employed to obtain the target PCR fragments.

### 2.4. Cloning and Characterization of the Full-Length Hsp cDNAs of the Giant River Prawn

A QIAquick^®^ gel extraction kit (Qiagen^®^) (Germantown, MD, USA) was used to purify PCR products from the above step, which were further subjected to the subsequent cloning process. All cDNA cloning steps and sequencing were carefully conducted following the description in a previous report [27]. The obtained cDNA sequences were searched for homology with other known sequences available in the GenBank database (http://www.ncbi.nlm.nih.gov) using the BLASTX and BLASTN programs. The 5′ RACE sequences and partial cDNAs of their Hsp counterpart clones were aligned to create the full-length cDNAs of all target genes. The full-length cDNAs of the *hsp40* and *hsp90* genes were searched for homology with sequences available in the GenBank database (NCBI, http://www.ncbi.nlm.nih.gov) using the BLASTX and BLASTN programs. The open reading frame (ORF) and the 5′ and 3′ untranslated regions (UTRs) of these cDNAs were analyzed using ORF Finder (Open Reading Frame Finder, http://www.ncbi.nlm.nih.gov/gorf/gorf.html). The full-length giant river prawn *hsp40* and *hsp90* cDNAs were further analyzed for the presence of a leader peptide using the DAS transmembrane prediction program (http://www.sbc.su.se/~miklos/DAS). The Simple Modular Architecture Research Tool program (http://smart.cmbl-heidelberg.de/) was used to predict the amino acid motif forms. The theoretical isoelectric point (p*I*) and molecular weight were computed by the compute p*I*/Mw tool program (http://www.expasy.org/tools/pi_tools.html). The deduced amino acid sequences were compared with those of other known *hsp* genes of various animal species using CLUSTAL W (http://ebi.ac.uk/Tools/clustalw/index.html) [28]. The homology of both the nucleotide and amino acid sequences was also determined using Matrix Global Alignment Tool (MatGAT) version 2.02 (http://bitincka.com/ledion/matgat/).

### 2.5. Phylogenetic Analysis of hsp Genes of Various Animal Species

Phylogenetic trees were generated using the deduced amino acid sequences encoded by giant river prawn *Hsp* genes and additional Hsp sequences of other species obtained from the GenBank database (see phylogenetic trees below). Closely and distantly related Hsp sequences from a wide range of eukaryotic (invertebrates, vertebrates, plants, fungi, protozoa) and prokaryotic species were used in the analysis. All analyzed leader sequences were excluded using the DAS transmembrane prediction server. All sequences were aligned by the CLUSTAL W program [28], and the phylogenetic tree was constructed using the neighbor-joining method implemented in the MEGA software version 6 [29]. Unrelated human Hsp sequences were used as outgroups. The reliability of the tree topology was assessed by 1000 bootstraps. 

### 2.6. Distribution of Giant River Prawn hsp Genes in Various Tissues of Healthy Prawns

#### 2.6.1. First-Strand cDNA Construction

One microgram of total RNA from each tissue that was previously prepared from healthy prawns (13 different tissues) was separately synthesized to generate first-strand cDNA by a RevertAid First-Strand cDNA Synthesis Kit (Fermentas, Waltham, MA, USA).

#### 2.6.2. Quantitative Real-Time RT-PCR (qRT-PCR)

To determine the gene expression levels of the *hsp40* and *hsp90* genes of giant river prawns in 13 different tissues, qRT-PCR was performed. In this experiment, one microliter of the first-strand cDNA from each tissue of 2 healthy prawns was subjected to qRT-PCR analysis, which was applied using an Mx3005P real-time PCR system (Stratagene, La Jolla, CA, USA) equipped with analytical software version 4.0 and Brilliant III SYBR Green qPCR Master Mix (Stratagene, La Jolla, CA, USA) according to the manufacturer’s recommended protocol. The qRT-PCR reaction was performed in a final volume of 18 µL containing 1 µL of the first-strand cDNA, 10 µL of 2x SYBR Green qPCR Master Mix, 5 µL of distilled water and 1 µL of each specific primer pair, including the RTF_GFP_hsp40 and RTR_GFP_hsp40 primers and the RTF_GFP_hsp90 and RTR_GFP_hsp90 primers for the *hsp40* and *hsp90* gene amplification, respectively (Table 1). The expression levels of the *hsp40* and *hsp90* genes in each sample from different tissues were normalized relative to the expression of the *β-actin* gene determined using the RTB-actin F and RTB-actin R primers (Table 1). The PCR conditions and analysis followed the methods described by Toe et al. [27].

The threshold cycle (Ct) of *hsp* and *β-actin* genes was measured, and a standard curve was used to determine their starting copy number. The relative copy number of the target mRNA was calculated according to the 2^−ΔΔCt^ method [30]. The Ct value difference (ΔCt) between *hsp* and *β-actin* mRNAs of each reaction was used to normalize the level of total RNA.

### 2.7. Expression Profiling of the Giant River Prawn hsp40 and hsp90 mRNAs under Aeromonas hydrophila Induction Determined Using qRT-PCR

#### 2.7.1. Experimental Animals

Three different groups of thirty healthy prawns, as described in Section 2.1, were randomly picked to maintain in a 1000 L fiberglass tank containing clean and fully oxygenated water for acclimatization for 7 days. During this time, the experimental prawns were fed twice per day with commercial feed at 3–5% body weight.

#### 2.7.2. *A. Hydrophila* Preparation

An *A. hydrophila* virulent strain (AQAH0101) was obtained from the Laboratory of Aquatic Animal Health Management, Department of Aquaculture, Faculty of Fisheries, Kasetsart University, Bangkok, Thailand. A single colony of *A. hydrophila* was carefully inoculated in 15 mL of tryptic soy broth (TSB, Merck, Kenilworth, NJ, USA) and incubated in a shaker water bath at 30 °C for 24 h. The bacterial suspension was centrifuged at 800× g for 15 min to harvest bacterial cells and washed twice with sterile 0.85% NaCl. The bacterial pellet was resuspended in sterile 0.85% NaCl. The concentrations of *A. hydrophila* were adjusted to 1 × 10^9^ CFU/mL with an optical density of 0.1 at 540 nm to obtain an original stock that was subsequently diluted with 0.85% NaCl to obtain another final concentration of 1 × 10^7^ CFU/mL.

#### 2.7.3. Induction Experiment

In the first and second experimental groups, every prawn in each group was intramuscularly injected with 0.1 mL of a bacterial suspension containing 1 × 10^7^ or 1 × 10^9^ CFU/mL of previously prepared *A. hydrophila* respectively, in the muscle under the area between the 3rd and 4th abdominal segments. In the last group, all prawns were similarly injected with 0.1 mL of 0.85% NaCl via the same route. After that, all injected prawns were maintained in their tanks under the same conditions as during the acclimatization step. During this time, behaviors and mortality were recorded daily.

#### 2.7.4. Total RNA Extraction and First-Strand cDNA Synthesis

Initially, 3 prawns from each group were sacrificed to collect the hemocytes, gills and hepatopancreas. After injection, the three target tissues were separately collected from 3 prawns from each group at 0, 3, 6, 12, 24, 48 and 96 h. Total RNA and first-strand cDNA of these tissues from the 3 prawns in each group at different times were extracted and synthesized respectively, with the same method as described above.

#### 2.7.5. qRT-PCR Analysis

One microliter of first-strand cDNA from each tissue of the control and infected prawns from Section 2.7.4 at different time points was subjected to qRT-PCR analysis, which was conducted as described above.

#### 2.7.6. Statistical Analysis

The relative copy number of each *hsp40* and *hsp90* mRNA among all tested groups of giant river prawns induced with *A. hydrophila* at different time intervals was calculated using the expression at 0 h in each tissue of prawns as an initial calibrator. Relative expression ratios of *hsp* genes were then statistically tested by one-way analysis of variance (ANOVA). The significance of the mean relative expression ratios was compared using Duncan’s new multiple range test (DMRT) at a 95% confidence level.

### 2.8. Expression Analyses of the Giant River Prawn hsp40 and hsp90 mRNAs in Response to Heat and Cold Shock Stimulation Determined by qRT-PCR

#### 2.8.1. Experimental Animals and Temperature Controls

Ninety prawns from Section 2.1 were randomly chosen to maintain in a 1000 L fiberglass tank containing clean and fully oxygenated river water for acclimatization for 7 days. During this period, the prawns were fed twice per day with commercial feed at 5% body weight, and 20% of the water was exchanged every day. The water temperature was maintained at 30.0 ± 0.4°C using a heater controlling system (HOPAR^TM^ K-339, Chosion, China). At day 8, 24 prawns each were further randomly moved to 3 separate new tanks. The first and second tanks were stabilized at temperatures of 30.0 ± 0.4 and 35.0 ± 0.7 °C respectively, using a heater controlling system (HOPAR^TM^ K-339, Chosion). In the last tank, the prawns were maintained in an air temperature conditioning room, which was previously set to 25.0 ± 0.4°C. All treated prawns were maintained in their tanks under the same conditions as those in the acclimatization step. Behaviors and mortality were recorded daily.

#### 2.8.2. qRT-PCR and Data Analyses

Prior to moving the experimental prawns, 3 of the 90 prawns were randomly sacrificed to collect the hemocytes, gills and hepatopancreas tissues. After moving to new tanks at 3, 6, 12, 24, 48 and 96 h, these three tissues were separately collected. Total RNA extraction and first-strand cDNA synthesis were conducted with the same method as described above. qRT-PCR analysis of the *hsp40* and *hsp90* genes in the hemocytes, gills and hepatopancreas in each temperature group at different times was carried out, and the relative copy number of *hsp* mRNAs among all tested groups of giant river prawns in response to different temperature levels at different times was calculated using the expression level at 0 h in each tissue of prawns as an initial calibrator. Subsequently, the relative expression ratios of the *hsp40* and *hsp90* genes were statistically tested, as described in Section 2.7.

### 2.9. Silencing Analysis and Effects of hsp40 and hsp90 Gene Knockdown on Disease Resistance

#### 2.9.1. Experimental Design

One thousand and five hundred juvenile giant river prawns (approximately 3 g) were acclimatized under laboratory conditions with the same practice as described above. Twenty prawns each were randomly maintained in 4 different 10 L glass tanks for 7 days.

#### 2.9.2. Double-Stranded RNA (dsRNA) Preparation

The gene-specific primers Mr-HSP40T7_F/Mr-HSP40T7_R, Mr-HSP90T7_F/Mr-HSP90T7_R and GFP_F/GFP_R (Table 1) were designed to obtain DNA templates containing a T7 promoter. These primers were used to amplify the cDNA from Section 2.6.1 with the same protocol, and the obtained PCR fragments were cloned into the pGEM T-easy vector, as previously described. Specific dsRNA for *Mr-hsp40, Mr-hsp90* and green fluorescent protein (GFP) was synthesized using the T7 RiboMAX™ Express RNAi System (Promega Corporation, Madison, WI, USA) and purified following the manufacturer’s protocol. Phosphate-buffered saline (PBS, pH 7.2) was used to dilute each dsRNA gene to reach a final concentration of 3 µg dsRNA/10 µL.

#### 2.9.3. Delivery of dsRNA and Expression Analysis of the hsp40 and hsp90 Genes

In this part of the experiment, all prawns prepared in the 4 tanks in Section 2.9.1 were intramuscularly injected under different conditions, as follows: in the first, second and third tanks, prawns were separately injected with 10 µL of PBS, PBS+3 µg of *Mr-hsp40* dsRNA, PBS+3 µg of *Mr-hsp90* dsRNA and PBS+3 µg of GFP dsRNA using a 10 mL microinjector syringe at the 3rd abdominal segment. After injection, at 0, 3, 6, 12, 24 and 48 h, gill, muscle and hepatopancreas samples of 3 prawns in each group were collected. Total RNA and first-strand cDNA were extracted and synthesized respectively, with the methods described above. qRT-PCR analyses of the *hsp40* and *hsp90* genes at each time were conducted with the same procedures described in Section 2.7.6.

#### 2.9.4. Effects of hsp40 and hsp90 Gene Silencing on High-Temperature Stress

Ten prawns, each stocked in Section 2.9.1, were placed in fifteen different 150 L glass tanks for seven days in a static water temperature at 30 ± 1.2 °C. After that, every prawn in tanks 1–3 (PBS control), 4–6 (GFP control group, normal temperature), 7–9 (GFP control group, high temperature), 10–12 (*hsp40* knockdown, high temperature) and 13–15 (*hsp90* knockdown, high temperature) was injected with 10 µL of PBS+3 µg of GFP dsRNA, PBS+3 µg of GFP dsRNA, PBS+3 µg of *Mr-hsp40* dsRNA and PBS+3 µg of *Mr-hsp90* dsRNA respectively, with the same methods as described above. During this experiment, the water temperature in tanks 7–15 was maintained at 35 ± 1.8 °C using the same equipment as in Section 2.8.1. After injection, prawns in each group were kept in their acclimatized tanks. Behavior and mortality were recorded at 0, 3, 6, 12, 24 and 48 h.

#### 2.9.5. Effects of hsp40 and hsp90 Gene Silencing on *A. hydrophila* Infection

Ten prawns, each stocked in Section 2.9.1, were placed in fifteen different 10 L glass tanks for seven days at a static water temperature of 30 ± 1.2 °C. The experimental design of the dsRNA injection was similar to that described in Section 2.9.4. An *A. hydrophila* solution was prepared under the same conditions as described in Section 2.7.2 to obtain a final concentration of 1 × 10^7^ CFU/mL. After dsRNA induction for 2 h, 20 µL of 1 × 10^7^ CFU/mL *A. hydrophila* in PBS was injected at the 3rd abdominal segment on the other side from the first injection, and the prawns in each group were kept in their acclimatized tanks. All data were recorded as described in Section 2.9.4.

### 2.10. Data Analysis

The relative expression of the *Mr-hsp40* and *Mr-hsp90* genes in Section 2.9.3 and cumulative mortality in Section 2.9.4 and Section 2.9.5 of each prawn group at different time points were statistically tested, as described in Section 2.7.6.

## 3. Results

### 3.1. Characterization and Sequence Analysis of Mr-hsp40 and Mr-hsp90

Based on the results of the present study, two EST clones containing the partial cDNAs P2714 and BG21346, which encode the giant river prawn *hsp40* and *hsp90* genes and their 5′-RACE sequence fragment respectively, were assembled to recover the full-length cDNA. The full-length giant river prawn cDNAs of the *hsp40* and *hsp90* genes, called “*Mr-hsp40* and *Mr-hsp90*” (GenBank database accession nos. KM081678 and KM081679, respectively), were recovered.

The *Mr-hsp*40 cDNA sequence is 2274 bp and contains a 5′ UTR of 79 bp, a 3′ UTR with a polyA tail of 1004 bp and an ORF of 1191 bp, encoding 396 amino acid residues. *Mr-hsp*40 has no signal peptide. The molecular weight and p*I* of Mr-hsp40 are 45 kDa and 6.11, respectively. Its amino acid structure presents four domains defining the *hsp40* type I protein family signature, consisting of an N-terminal conserved domain or a J domain (aa 5–60) that shows a conserved HPD tripeptide, a G/F domain (aa 63–96) with a conserved DIF motif, a CR domain or central domain (aa 122–258) with three CXXCXGXG repeats (where X is any amino acid) and a CTD (aa 236–338). The CQTS motif of *Mr-hsp*40 indicates that it is protein farnesylation of *DnaJ* (*hsp40*) subfamily A member 1 (Figure 1).

The cDNA sequence of *Mr-hsp*90 is 2721 bp, composed of a 5′ UTR of 113 bp, a 3′ UTR with a polyA tail of 424 bp and an ORF of 2184 bp, encoding 727 amino acid residues that contain a signal peptide (aa 1–27) and the mature protein. The molecular weight and p*I* of Mr-hsp90 are 84 kDa and 4.87, respectively. The Mr-hsp90 amino acid sequence has an ATP-binding domain (aa 34–188) with a conserved GXXGXG motif and five conserved amino acid motifs defining the Hsp90 protein family signatures (N^33^KEIFLRELISNSPDALDKIR^53^, L^100^GTIAKSGT^108^, I^124^GQFGVGFYSAYLIAD^139^, I^355^KLYVRRVFI^364^ and G^381^VVDSEDLPLNISRE^367^). The MEEVD motif at the C-terminus of this protein is present in all *hsp90* sequences in the cytoplasm (Figure 2).

Multiple alignment sequence analysis with other Hsp40 species showed that the Mr-hsp40 amino acid structure is highly similar among *hsp40* type I proteins in protozoa and vertebrates in the N-terminal conserved domain, G/F domain, CR domain and CTD, especially the HPD tripeptide motif (H^34^PD^36^), DIF motif (D^87^IF^89^) and three CXXCXGXG motifs (C^135^SKCEGQG^142^, C^150^PTCRGTG^157^ and C^177^SDCRGQG^184^). In contrast, Mr-hsp40 contains a CQTS motif that is found in vertebrates and crustaceans (Figure 3). The Mr-hsp90 amino acid sequence is extremely similar to those of other species in five conserved amino acid motifs and the consensus sequence MEEVD at the C-terminus. In addition, Mr-hsp90 has three N-linked glycosylation sites, containing two sites (N^285^KT^287^ and N^393^IS^395^) found in other *hsp90* species and one site (N^434^FS^436^) found only in the Hsp90 proteins of crustaceans. Moreover, Mr-hsp90 has the addition of amino acid sequence D^243^KDKEKE^249^ in the charged linker (Figure 4).

### 3.2. Homological Analysis of the Nucleotide and Amino Acid Sequences and Phylogenetic Tree Analysis of Mr-hsp40 and Mr-hsp90

Comparisons of nucleotide identity and amino acid identity and similarity revealed that *Mr-hsp40* displays 50.8–77.9% nucleotide identity, 37.7–87.4% amino acid identity and 57.7–95.2% amino acid similarity with other *hsp40* type I genes and proteins in eukaryotic animals. *Mr-hsp40* has high identity and similarity with invertebrate *hsp40* gene and proteins, especially in those in kuruma shrimp (*Marsupenaeus japonicus*), at 77.9% nucleotide identity, 87.4% amino acid identity and 95.2% amino acid similarity (Appendix A). *Mr-hsp90* exhibits 50.6–85.8% nucleotide identity, 34.8–90.9% amino acid identity and 58.3–96.4% amino acid similarity to other *hsp90s* in all organisms. *Mr-hsp90* shows the highest amino acid identity and similarity with crustacean proteins and particular similarity to that of the oriental river prawn (*Macrobrachium nipponense*), at 90.9% identity and 96.4% similarity. Additionally, the *Mr-hsp90* sequence is similar to that of *M. rosenbergii hsp90* (HG001457) determined in another study, with amino acid scores of 80.9% identity and 90.1% similarity (Appendix A).

Phylogenetic tree analysis established the evolutionary relationship of *M. rosenbergii hsps* with other known *hsp40* and *hsp90* genes found in the GenBank database. There were 77 *hsps* from different eukaryotes (vertebrates, invertebrates, plants, fungi and protozoa) and prokaryotes in the phylogenetic tree, which indicated that these *hsps* were separated into three main clusters consisting of the *hsp40* and *hsp90* families (Figure 5 and Figure 6, respectively). The *hsp40* family had four subclusters: hsp40 type I, type II, type III and prokaryote, in which *Mr-hsp40* was clustered in the invertebrate *hsp40* type I group (Figure 5). The *hsp90* clustering indicated that *Mr-hsp90* is closely related to *hsp90* in crustaceans, especially that in *M. nipponense.* In contrast, *M. rosenbergii hsp90* (HG001457), which was previously discovered, is closely related to *Exopalaemon carinicauda hsp90* (Figure 6).

### 3.3. Expression Analysis of the Giant River Prawn hsp40 and hsp90 Genes

The expression patterns of the *Mr-hsp40* and *Mr-hsp90* genes in every examined organ of normal prawns determined by qRT-PCR demonstrated that *Mr-hsp40* was mildly expressed in the testis, gills and hepatopancreas, with expression levels of 3.94 ± 0.16- to 6.75 ± 0.19-fold compared to those in the eyestalk (Figure 7), while *Mr-hsp90* was ubiquitously expressed, with the highest expression observed in the ovary in females (37.64 ± 2.38) and moderate expression in the gills, hindgut, hemocytes, hepatopancreas, midgut and thoracic ganglia, with expression levels of 4.45 ± 0.28- to 12.08 ± 1.15-fold compared to those in the eyestalk (Figure 7).

### 3.4. Expression Analyses of Giant River Prawn hsp40 and hsp90 mRNAs in Response to Stress Conditions

#### 3.4.1. Expression Analyses of Mr-hsp40 and Mr-hsp90

qRT-PCR was used to quantify the *Mr-hsp40* and *Mr-hsp*90 mRNAs in the gills, hepatopancreas and hemocytes after injection with 1 × 10^7^ or 1 × 10^9^ CFU/mL *A. hydrophila* at different time points. No mortality was observed during this experiment, but some lethargic prawns could be found at hI 24–48. The *Mr-hsp40* expression level in the gills of prawns injected with 1 × 10^9^ CFU/mL *A. hydrophila* was significantly (*p* < 0.05) upregulated at 3 (1.48-fold) and 12 h after injection (hI) (2.52-fold), after which the peak expression levels of prawns injected with the two concentrations compared with those in the control prawns were significantly upregulated by 4.04- and 2.70-fold at 48 hI (Figure 8A). In the hepatopancreas, *Mr-hsp40* expression levels in prawns injected with the two concentrations were significantly upregulated at 6 hI (1.55- and 3.39-fold) and highly expressed again in prawns injected with 1 × 10^9^ CFU/mL *A. hydrophila* at only 48 hI (2.51-fold; Figure 8B). The *Mr-hsp40* expression patterns in hemocytes showed significant upregulation in prawns injected with 10^7^ CFU/mL *A. hydrophila* at 12 (5.87-fold) and 96 hI (4.20-fold) and in prawns injected with 10^9^ CFU/mL *A. hydrophila* at 3 (1.73-fold), 12 (5.11-fold), 24 (5.14-fold) and 96 hI (5.65-fold; Figure 8C).

The *Mr-hsp90* expression level in the gills of prawns injected with 10^7^ CFU/mL *A. hydrophila* was significantly upregulated at 3 (2.48-fold), 6 (1.83-fold), 12 (1.62-fold), 48 (1.92-fold) and 96 hI (1.99-fold), but was upregulated only at 3 hI in prawns injected with 1 × 10^9^ CFU/mL *A. hydrophila* (3.80-fold; Figure 9A). Regarding expression level in the hepatopancreas, *Mr-hsp90* was highly upregulated in prawns injected with 10^7^ CFU/mL *A. hydrophila* at 3 (7.11-fold), 12 (6.80-fold) and 96 hI (1.42-fold), significantly upregulated in prawns injected with 1 × 10^9^ CFU/mL *A. hydrophila* at 6 hI (6.40-fold) and slightly upregulated in prawns injected with 1 × 10^9^ CFU/mL *A. hydrophila* at 24 (1.77-fold) and 48 hI (3.15-fold; Figure 9B). In hemocytes, *Mr-hsp90* expression patterns showed that in prawns injected with the two concentrations, *hsp90* was significantly highly upregulated at 96 hI (89.92- and 128.52-fold; Figure 9C).

#### 3.4.2. Expression Analyses of Giant River Prawn Mr-hsp40 and Mr-hsp90 mRNAs in Response to Heat and Cold Shocks

Interestingly, all applied temperature levels did not cause mortality through the end of the experimental periods. However, at 12–96 h after temperature change (hT), experimental prawn exposed to low temperature (25 °C) showed lethargic activity, while high temperature (35 °C) induced early vigorous swimming effects at the first 12 hT and later tempered lethargic effects until 96 hT. Furthermore, the physical appearance of colored hemolymph at 48–96 hT exhibited a pale turquoise coloration at 30 °C, light peach coloration at 35 °C and moderate blue coloration at 25 °C (Figure 10). qRT-PCR was applied to determine the mRNA expression patterns of the *Mr-hsp40* and *Mr-hsp*90 genes in the tested tissues at different time points after heat–cold shock (Figure 11 and Figure 12, respectively). *Mr-hsp40* expression levels in the gills at 35 and 25 °C were significantly (*p* < 0.05) downregulated at 6 hT (0.21- and 0.29-fold) and upregulated at 48 hT (7.02- and 6.60-fold; Figure 11A). In the hepatopancreas, *Mr-hsp40* expression was significantly upregulated at 12 (3.79-fold) and 24 hT (3.79-fold) at 35 °C, and peaked at 96 hT (20.76-fold) at 25 °C (Figure 11B). In hemocytes, *Mr-hsp40* expression was significantly upregulated at 6 (1.97-fold) and 48 hT (3.24-fold) at 35 °C, and subsequently significantly upregulated at 96 hT (2.39-fold) at 25 °C (Figure 11C).

*Mr-hsp90* expression in the gills at 35 °C was initially significantly upregulated at almost all experimental periods and, in the 25 °C group, was significantly downregulated, and upregulated only at 48 hT (5.70-fold; Figure 12A). In the hepatopancreas, *Mr-hsp90* expression at 35 and 25 °C was significantly downregulated 0.15- and 0.23-fold at 6 hT and then upregulated 12.05- and 12.69-fold at 12 hT, respectively (Figure 12B). In hemocytes, mRNA upregulation of *Mr-hsp90* was observed at approximately 35 °C at different time points and peaked at 28.65-fold at 3 hT, and downregulation of *Mr-hsp90* was observed at 25 °C, at 6 (0.53-fold), 24 (0.18-fold) and 96 hT (0.35-fold; Figure 12C).

### 3.5. Gene Knockdown Analysis

In this study, *Mr-hsp40* and *Mr-hsp90* were clearly silenced in all tested tissues (Figure 13A,B for the gills, Figure 13C,D for the hepatopancreas and Figure 13E,F for muscle). Based on the data obtained from this experiment, ds*Mr-hsp40* and ds*Mr-hsp90* were effective in causing the degradation of their target mRNA counterparts. The silencing effect of these 2 dsRNAs obviously demonstrated their effective functions in initially knocking down mRNAs of the *Mr-hsp40* and *Mr-hsp90* genes at 12–24 h after treatment. Significantly decreased mRNA levels of these 2 genes in the silencing groups were observed from when they were strongly silenced through the end of the experiments at 96 h (*p* < 0.05) compared to those in the PBS- and dsGFP-injected groups. Particularly for the gills and hepatopancreas, where the *Mr-hsp90* gene was expressed in high amounts with log copy number/100 ng cDNA of 6.41 and 5.24 respectively, the silencing effects on this gene were still significantly effective (Figure 13B and Figure 3D) (*p* < 0.05).

### 3.6. Effects of Gene Knockdown on the Response to Temperature Stress and A. hydrophila Injection

Under high-temperature conditions, prawns injected with dsGFP, ds*Mr-hsp40* and ds*Mr-hsp90* showed rapid mortality during 6–24 hI. The highest mortality was recorded at 48 hI, at 23.33% ± 5.77%, 66.67% ± 11.55% and 96.67% ± 5.77%, which was significantly different from that of PBS-injected prawns, which had no mortality (0.0% ± 0.0%) during this period (*p* < 0.05) (Figure 14A).

Additionally, it was shown that the prawn groups injected with ds*Mr-hsp40* and ds*Mr-hsp90* exhibited very rapid mortality after injection with *A. hydrophila* during 6–12 hI. The mortality of these prawn groups reached 100.0% at 12 hI, while the mortality of the GFP control (PBS-injected group) and GFP (*A. hydrophila*-injected group) groups gradually increased from 6 to 48 hI. At 48 hI, the mortality of the GFP control and GFP groups was 16.67% ± 11.55% and 53.33% ± 15.28% respectively, which was significantly lower than that of the ds*Mr-hsp40* and ds*Mr-hsp90* knockdown groups, which showed 100.0% mortality (*p* < 0.05) (Figure 14B).

## 4. Discussion

Generally, Hsps have been classified as important biomolecules that play critical roles in the response to cellular stressors and have been discovered in many economically important aquatic species. In this study, new full-length cDNAs of two Hsps were isolated from giant river prawns and characterized.

Structural analysis revealed that the *Mr-hsp40* amino acid structure has components similar to those found in other organisms. It importantly contains the J domain, which controls the ATP hydrolytic cycle of Hsp70s, and the CT domain, which carries and transfers denatured proteins to their chaperone partners [13,15,31]. Typically, the J domain has four helices and a loop position between helices II and III that recognizes a highly conserved HPD tripeptide motif [14]. The amino acid sequence of *Mr-hsp40* clearly groups within the Hsp40 type I cluster, is highly homologous to that of *hsp40* of *M. japonicus* [32] and possesses the specific characteristics of Hsp70s needed to efficiently perform protein folding and especially manage normal cellular functions [31]. In addition, the C-terminus contains a farnesylated protein (CQTS form), which was specifically classified in the Hsp40 subfamily A member 1 (DnaJA1) group and exhibits various properties in cellular systems [33].

For Mr-hsp90, the amino acid structure at the N-terminus shows a leader peptide that promotes transformation into the extracellular space; however, these peptides become amphiphilic helices to functionally maintain cellular systems under normal conditions [34]. Mr-hsp90 also contains five conserved amino acid Hsp90 protein family signatures and an ATP-binding domain, which is important for controlling the crucial mechanisms of Hsp90s [35,36]. In addition, the C-terminus of Mr-hsp90 was found to carry the consensus MEEVD motif, indicating that this molecule is a member of the cytosolic Hsp90 family [37]. It is also called a tetratricopeptide repeat that can bind with other molecular chaperones, such as other Hsp90s, Hsp70 and the proteasome, to form multiprotein complexes [36]. Moreover, Mr-hsp90 can be considered a glycoprotein because it has N-linked glycosylation sites used in binding with oligosaccharides or glycans [38]. Additionally, the Mr-hsp90 sequence shows an amino acid addition in the charged linker that may enhance Hsp90 function and restore the flexibility and adaptability of its structure to detect and process targeted client proteins [39].

Homology comparisons of nucleotide identity and amino acid identity and similarity revealed that *Mr-hsp40* and *Mr-hsp90* have the highest percent identity and similarity scores to those of crustaceans, supporting their being highly conserved genes among organisms. Interestingly, *Mr-hsp90* displays an amino acid similarity of 90.1% to the previously described *M. rosenbergii hsp90* (HG001457), which has an N-region 386 amino acids shorter than Mr-hsp90 in our study. This result suggests that another *hsp90* existed in the giant river prawn. This information supports the report of Pantzartzi [40], who discovered that an exclusive Hsp90 located in the cytoplasm can be encoded by one or two genes in invertebrates.

Phylogenetic tree analysis confirmed the Hsp homology results. *Mr-hsp40* and *Mr-hsp90* were clustered in their Hsp families and highly evolutionarily related to crustacean groups, which was a similar result to that of a previous study [41]. Moreover, *Mr-hsp40* grouped in the Hsp40 type I cluster that was clearly divided from the Hsp40 type II and III groups of vertebrates [42], which proves that prawn Hsp40 evolved from the orthologous gene in vertebrates. The evolution of *Mr-hsp90* was determined in a previous study in *P. trituberculatus* [43], which demonstrated that two divergences of cytosolic Hsp90 could have occurred in crustaceans.

Expression analysis of the *Mr-hsp40* and *Mr-hsp*90 genes in various tissues indicated that *Mr-hsp40* was highly expressed in the testis, which is similar to DnaJA1 expression in mammals. In higher vertebrates, one or two Hsp40s, preferably DnaJA1, are regulated by androgen receptor signaling that is involved in maintaining spermiogenesis [44]. In contrast*, Mr-hsp90* expression was observed in almost all tested tissues, especially the ovary, which exhibited the highest expression level. In general, Hsp90 makes up 1–2% of total cytosolic proteins under nonstress conditions and performs critical roles in protein synthesis and important functions to regulate the expression of other *Hsp* transcripts [45,46]. Moreover, ovarian development in vertebrates requires Hsp90 to bind with the estrogen receptor to release estrogen hormone, and these mechanisms play an essential role in maintaining vitellogenin synthesis in the liver [47,48]. In contrast, crustaceans have no estrogen hormone; however, they can produce vitellogenin [49]. Recent studies indicate that Hsp90 expression levels in crustacean ovaries are an important effector of ovarian development through regulating vitellogenin synthesis [50,51].

*A. hydrophila* is an important problem in the aquaculture industry that can infect and cause harmful diseases in a wide variety of economically important species, including *M. rosenbergii*. Generally, *A. hydrophila* is also an opportunistic pathogen that causes disease, manifesting in clinical signs when hosts become stressed or coinfected with other pathogens [52]. The report of Abdolnabi et al. [53] demonstrated that surviving *A. hydrophila*-injected prawns had many histopathological changes found in several organs that included focal necrosis, hemolytic infiltration, hyperplasia, muscular damage and mild or massive hemocyte reaction. Under infection conditions, Hsps have an important role in maintaining normal cellular function by protein stabilization and refolding, contributing to the primary immune response against pathogens [54]. In addition, some Hsps, including Hsp27, Hsp60, Hsp70, Hsp90 and Grp78, can be stimulated and transferred into the extracellular space [55,56,57,58]. Likewise, necrosis is caused by the toxicities of pathogens and is also one of the pathways by which extracellular Hsps are released for the detection of denatured proteins or pathogens in the cellular immune response [59].

In this study, analysis of the *Mr-hsp40* and *Mr-hsp90* response in the gills, hepatopancreas and hemocytes revealed that *Mr-hsp90* exhibited a stronger response. The high expression level of *Mr-hsp90* in the initial periods after injection showed patterns similar to that of Hsp90 in *P. monodon* [60] and *Labeo rohita* [61]. Under stressful conditions, Hsp90 is an early molecular chaperone that facilitates the repair of misfolded proteins and prevents protein aggregation [12,62]. Moreover, Hsp90 is able to directly bind with the lipopolysaccharide of Gram-negative bacterial membranes, such as *A. hydrophila* and *Vibrio* sp. [63,64]. Furthermore, Hsp90 interacts with a CD91 receptor on the cell surface of phagocytes and can further induce the activities of innate immune cells to take up targeted contaminants [65,66].

Downregulation of these two genes in the target organs at 24–96 hI was found in the gills and hepatopancreas of *A. hydrophila*-infected prawns. These expression patterns were similar to those in previous studies. For example, downregulation of Hsp90 at 48 h in the gills of *P. monodon* injected with *Vibrio harveyi* has also been observed [60], and downregulation of Hsp90 in *A. hydrophila*-infected rohu has been observed between 24 and 72 h in the liver [61]. The observed changes in expression may correlate with the amount of pathogen in the host and be downregulated by proteins and cells to resist disease, and then, the cells become quiescent. However, upregulation of these two Hsps was also observed here, and the highest peak expression of *Mr-hsp90* in hemocytes was observed at the last experimental period (96 hI). This result may be because prawn hemocytes can eventually detect invading pathogens and then release Hsp40 and Hsp90 molecules to stabilize denatured proteins or refold impaired proteins at the late stage of infection [25]. However, the mRNA expression level of *Mr-hsp90* had a different pattern than that in a previous report by Chaurasia et al. [39], who reported that *M. rosenbergii* Hsp90 (HG001457) was not significantly expressed at 3–6 h after *A. hydrophila* injection, suggesting that the *Mr-hsp90* gene in this study has different functions in response to *A. hydrophila*.

Temperature is a difficult factor to control on a wide scale, but it dramatically affects the maintenance of poikilotherms [67]. Increasing or decreasing temperature is an essential part of regulating physiological changes affecting the production period in aquaculture for manufacturing at market scale and producing new offspring per year. In addition, temperature has a certain effect on the health of aquatic organisms by controlling growth and metabolic and respiratory rates. Moreover, it also regulates the invasion and virulence of bacterial pathogens, causing a negative effect on fish and shellfish production [3,4,68]. In addition, sudden temperature change ranges over 3–5 °C threaten aquatic animals, causing stress, shock and subsequently, death [69]. Elevated temperature has a straightforward effect on aquaculture conditions by reducing dissolved oxygen content in water and increasing the toxicity of some water qualities, while aquatic living conditions require additional oxygen to respond to increased metabolic and respiratory rates [70]. In addition, heat shock conditions induce cellular stresses, including protein misfolding [71]. However, the differential expression of prawn Hsps indicated that they have an important role in the host against heat shock. Our results indicated that significant upregulation of *Mr-hsp90* occurred in the gills at initial periods to stabilize respiratory function and to increasingly absorb oxygen from higher-density water. Likewise, previous reports have shown similar patterns in the thermal tolerance of crustacean Hsps, including Hsp90 of *Fenneropenaeus chinensis* [72], Hsp90 of *P. monodon* [60], Hsp40 and Hsp90 of *M. japonicus* [32] and Hsp90 and Grp78 of *Eurytemora affinis* [73]. Non-significant expression of all tested Hsps was observed in the hepatopancreas at 3–6 h after treatment. In general, the upregulation levels of Hsps in the hepatopancreas of crustaceans have been found to respond to thermal changes more slowly than that in other tissues [74]. However, their expression, especially for *Mr-hsp90*, was highly upregulated at subsequent periods, which might be related to crustacean hyperglycemic hormone (CHH), which also controls homeostatic metabolism and hemolymph glucose levels [75]. Moreover, Hsp90 is also involved in binding with hormones to transfer their complexes from the hepatopancreas to targeted organs [51,76].

Thus, *Mr-hsp90* expression may assist hepatopancreatic function to maintain the CHH pathway. However, the hemolymph showed different colors in each group, which may correlate with the alteration in protein and hemolymph parameters. Several reports have indicated that increasing temperature is directly related to changing concentrations of blood components in prawns, including elevated total hemocyte counts and lactate, glucose and protein concentrations, major DNA damage [77], disturbance of ion homeostasis, such as that of Na^+^, K^+^ and Ca^2+^ [78], and elevated oxidative stress due to enhanced reactive oxygen species activity [64,79]. Due to this effect, *Mr-hsp40* in hemocytes was shown to be briefly upregulated at initial periods to protect the host cell by rebalancing protein synthesis and stabilizing Ca^2+^ homeostasis to resist oxidative stress [80], while *Mr-hsp90* was shown to have significantly higher expression at almost all times because its function is necessary for cellular processes at high temperatures by obstructing protein aggregation during folding or refolding [60,81].

On the other hand, lower temperature is known to strongly influence aquatic survival by decreasing metabolic and respiratory rates. In addition, long-term cold-water conditions had no effect on the prawn survival rate but reduced the growth rate and food conversion ratio [82]. Likewise, this study showed that non-expressed or suppressed mRNAs of prawn Hsps were present in all tested tissues because cold temperature may reduce Hsp levels by relieving cellular functions, including protein synthesis [67]. However, the high upregulation of *Mr-hsp90* might be correlated with negative activities of cellular functions. According to a previous study by Qiu et al. [83], the physiological and immunological changes in shrimp hemolymph under low-temperature stress had decreasing trends for total hemocyte count, total protein concentration and osmolality in the plasma, but oxidative stress, DNA damage, lipid peroxidation and changes in osmolality in *Litopenaeus vannamei* could be enhanced in a short period of time. However, *Mr-hsp40* expression was significantly upregulated at 48–98 hT, which may work together with other chaperones, such as Hsp70s. In addition, subtropical aquatic animals synthesize some Hsps, such as Hsp40 and Hsp70, to regulate ATP hydrolysis to maintain protein and hormone homeostasis and survive long-term cold stress [84,85].

In crustaceans, functional analysis of *hsp* gene expression via RNAi, especially for *hsp40* and *hsp90*, is very limited. In 2018, Chen et al. [86] reported silencing of *Lv*-*HSP40*, which is an Hsp 40 type I, similar to the observations for *Mr*-*hsp40* in this study. *Lv*-*HSP40* was initially silenced in the hepatopancreas and ovary at 12 h after induction. Interestingly, silencing of this gene was shown to strongly affect the gonadosomatic index and ovarian maturation of the target animals.

In our current study, *Mr*-*hsp40* and *Mr*-*hsp90* expression was successfully knocked down in the gills, hepatopancreas and muscle. The transcripts of these 2 genes were suppressed early at 12–24 h in all tested tissues after induction with strong and significant silencing, and still showed similar results until 96 h. Interestingly, when the RNAi-induced groups were exposed to both the high temperature and a pathogenic bacterium, mortality was observed early at 6 h. This indicates that the injected RNAi may enhance the other death-involving factors, make them more sensitive to these inducers, and even the two transcripts of tested *hsp* genes are not strongly silenced yet.

When these silenced prawns were placed under high-temperature conditions at 35 °C, silencing of *Mr*-*hsp90* more severely induced mortality than *Mr*-*hsp40* RNAi induction. This result suggested that *Mr-hsp90* is more crucial in the response to high-temperature conditions than *Mr-hsp40*. Additionally, when the *Mr*-*hsp40-* and *Mr*-*hsp90-*silenced prawns were injected with a pathogenic bacterium (*A. hydrophila*), the prawns in these 2 groups were more seriously sensitive to bacterial infection than those in the GFP control (no bacterium) and GFP groups, since they reached 100% mortality at 12 h after injection with pathogenic bacteria, while the GFP control and GFP prawns showed mortality of 16.67% and 53.33% respectively, at 96 h. The disruption of the *hsp* gene transcripts via knockdown techniques in this study may have effectively affected all crucial activities of these 2 genes [52,53,54,55,56,57,58,59,60,61,62,63,64,65,66,73,74,75,76,77,78,79,80,81], and this information strongly supports the important roles of these 2 genes in *A. hydrophila* infection in giant river prawns.

Based on the above information, our study provides evidence demonstrating significant crucial roles of *Mr*-*hsp40* and *Mr*-*hsp90* as effective biomolecules that are importantly involved in both heat stress and *A. hydrophila* invasion. During stress responses to these stressors, stabilization denatured proteins by Hsp90, and the co-function at the ATP-dependent site of the Hsp70-Hsp40 chaperone system to refold impaired proteins may be disrupted [25]. Additionally, disruption of these chaperones may seriously affect ERAD that cannot functionally destroy aberrant secreted proteins resulting from stress conditions. Additionally, the CFTR structural integrity, essential for the ERAD system, is not well-maintained and is appropriately degraded by Hsp90 and Hsp40, respectively [26]. Regarding the underlying mechanisms, further investigation for a clearer description is important. Unexpectedly, induction by RNAi in the GFP-injected groups resulted in some negative side effects, indicated by some mortality that occurred during RNAi induction in both the heat and bacterial treatment groups. Further experiments are also needed to clarify this phenomenon and optimize this approach.

In conclusion, the full-length cDNA of the *Mr-hsp40* and *Mr-hsp90* genes was successfully cloned and characterized. Characterization and sequence analysis revealed that *Mr-hsp40* and *Mr-hsp90* have conserved amino acid motifs characteristic of those family signatures. Multiple sequence alignment and comparisons of nucleotide and amino acid sequences of the *Mr-hsp40* and *Mr-hsp90* genes with those of other species showed very high percent similarity scores to those of invertebrates, supporting the highly conserved nature of these genes among organisms. Expression and distribution analysis of the *Mr-hsp40* and *Mr-hsp90* genes in various tissues of normal prawns by qRT-PCR revealed that *Mr-hsp40* was highly expressed in the testis and gills, while the *Mr-hsp90* gene was ubiquitously expressed, with high expression in the ovary and moderate expression in the gills, hindgut, hemocytes, hepatopancreas, midgut and thoracic ganglia. Expression and functional analysis of *Mr-hsp40* and *Mr-hsp90* in normal and various stress conditions in the gills, hepatopancreas and hemocytes showed that these genes are crucially involved in cellular stress responses to both temperature and infection stimuli.

## Figures and Tables

**Figure 1 biomolecules-11-01034-f001:**
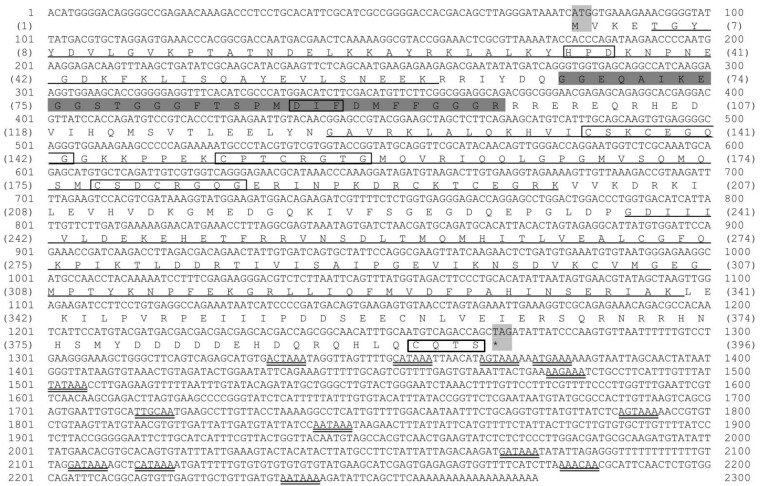
Nucleotide and deduced amino acid sequences of the *Mr*-*hsp40* gene. All conserved amino acid motifs defining the hsp40 family signature (HPD, DIF and CXXCXGXG) and the consensus sequence CQTS at the C-terminus are shown in the open box. The N-terminal conserved domain (J domain), the central domain (CRR domain) and the C-terminal domain (CTD) are underlined. The glycine/phenylalanine-rich region (G/F) domain is highlighted by a gray background. N-linked glycosylation sites are shown by points. Polyadenylation signal sites are double underlined.

**Figure 2 biomolecules-11-01034-f002:**
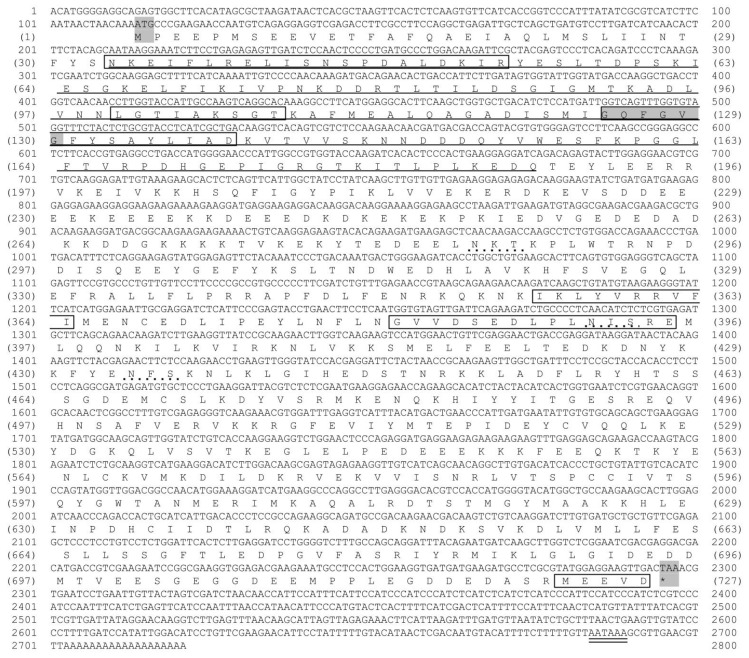
Nucleotide and deduced amino acid sequences of the Mr-hsp90 gene. All five conserved amino acid motifs defining the hsp90 family signature (NKEIFLRELISN[S/A]SDALDKIR, LGTIA[K/R]SGT, GQFGVGFYSA[Y/F]LVA[E/D], IKLYVRRVFI, and GVVDS[E/D]DLPLN[I/V]SRE) and the consensus sequence MEEVD at the C-terminus were present in the hsp90 sequence and are shown in the open box. The ATPase domain of hsp90 is underlined. N-linked glycosylation sites are shown by points. The polyadenylation signal site (AATAAA) is double underlined.

**Figure 3 biomolecules-11-01034-f003:**
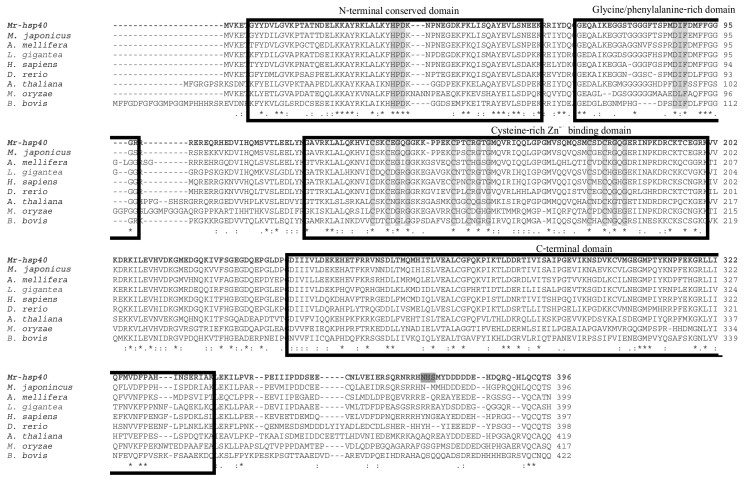
Multiple sequence alignments of *Mr-hsp40* with those of other species found in the GenBank database. *HSP40* family signature sequences and the consensus sequence “CQTS” at the C-terminus are shown in the open boxes. The symbols (*, : and .) in the CLUSTAL W alignment represent identical residues, conserved substitutions and semi-conserved substitutions, respectively. The accession numbers of the other species are shown in Figure 5.

**Figure 4 biomolecules-11-01034-f004:**
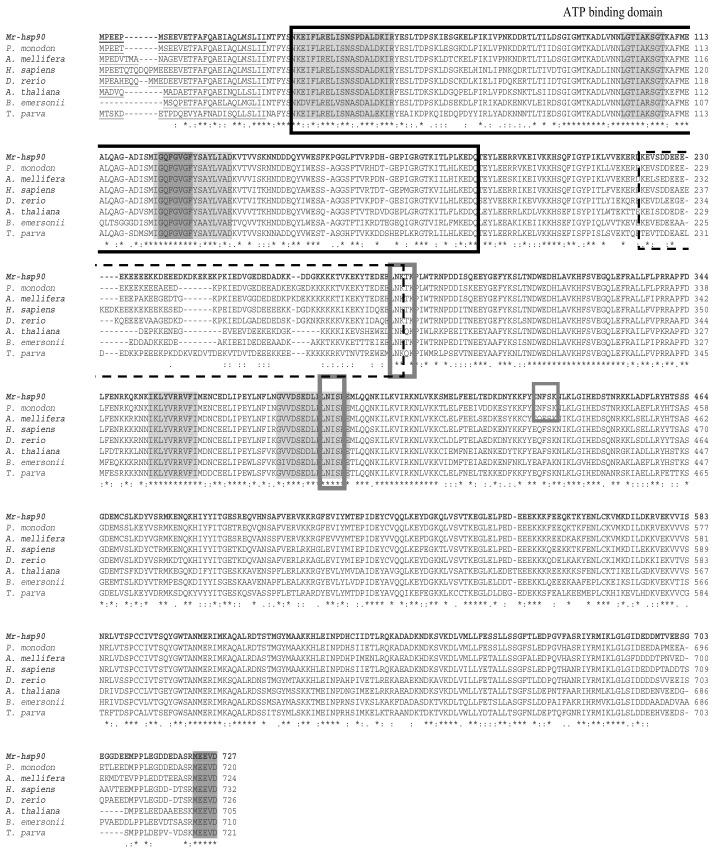
Multiple sequence alignments of *Mr-hsp90* with those of other species found in the GenBank database. *HSP90* family signature sequences and the consensus sequence “MEEVD” at the C-terminus are shaded with boxes. The symbols (*, : and .) in the CLUSTAL W alignment represent identical residues, conserved substitutions and semi-conserved substitutions, respectively. The accession numbers of the other species are shown in Figure 6.

**Figure 5 biomolecules-11-01034-f005:**
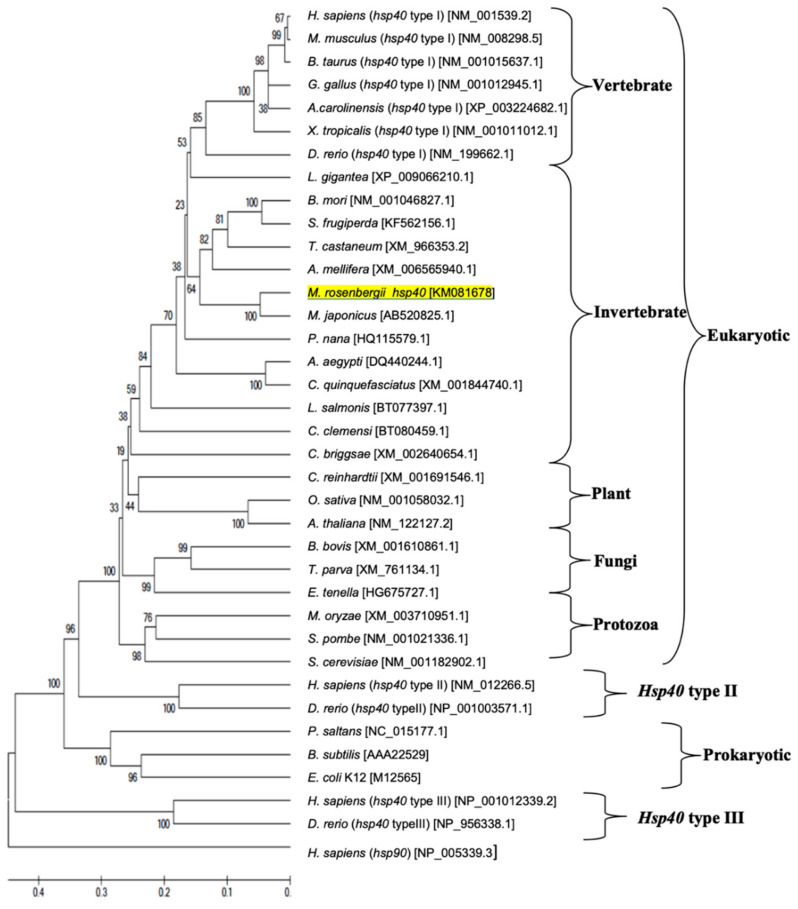
Phylogenetic tree showing the relationship of the *Hsp40* gene of the *Macrobrachium rosenbergii* amino acid sequence with those of other known species produced using the neighbor-joining method. Sequences used in the tree, followed by their GenBank accession number, are shown in Appendix A. *Mr-hsp40* is underlined.

**Figure 6 biomolecules-11-01034-f006:**
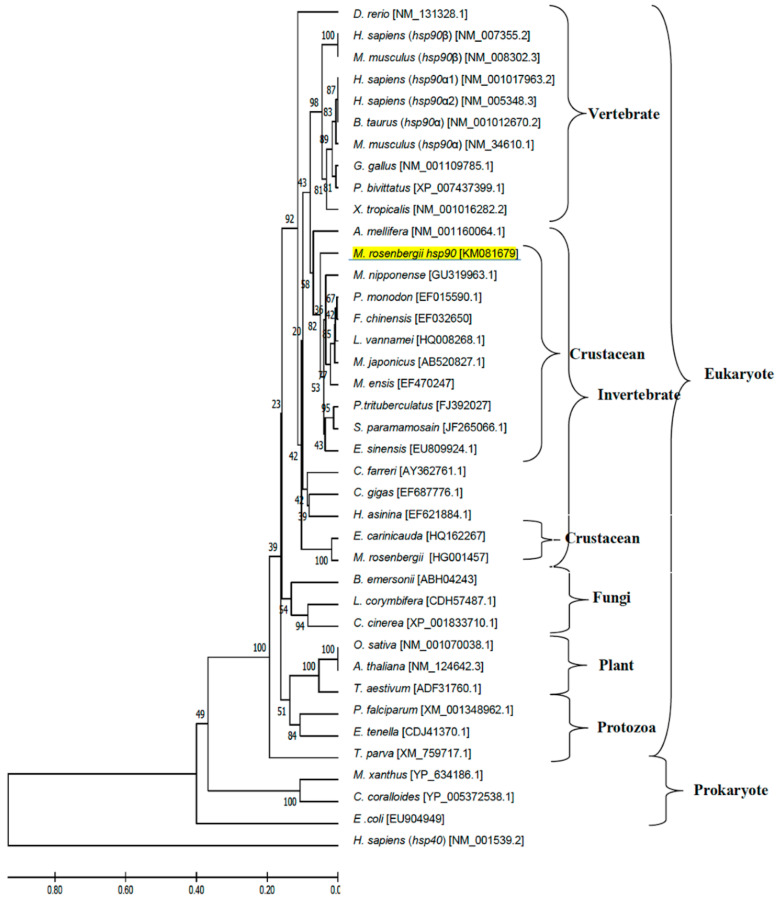
Phylogenetic tree showing the relationship of the *Hsp90* gene of the *Macrobrachium rosenbergii* amino acid sequence with those of other known species produced using the neighbor-joining method. Sequences used in the tree, followed by their GenBank accession number, are shown in Appendix A. *Mr-hsp90* is underlined.

**Figure 7 biomolecules-11-01034-f007:**
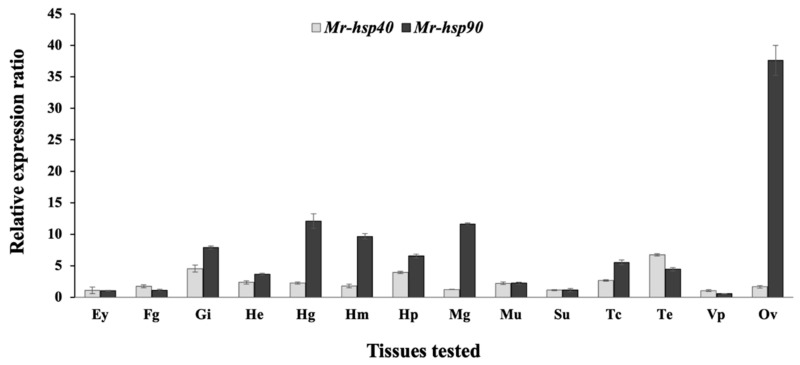
Expression analysis by qRT-PCR of the *Mr*-*hsp40* and *Mr*-*hsp90* genes. Ey = eyestalk, Fg = foregut, Gi = gills, He = heart, Hg = hindgut, Hm = hemocytes, Hp = hepatopancreas, Mg = midgut, Mu = muscle, Su = subcuticular epithelium, Tc = thoracic ganglia, Te = testis, Vp = *vas* deferens and Ov = ovary.

**Figure 8 biomolecules-11-01034-f008:**
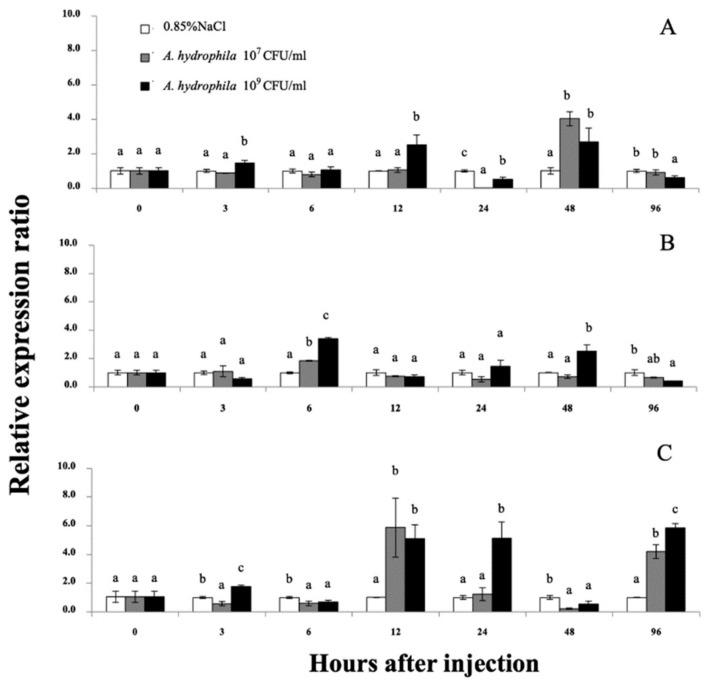
The *Mr-hsp40* mRNA expression levels relative to *β-actin* levels analyzed by qRT-PCR in the gills (**A**), hepatopancreas (**B**), and hemocytes (**C**) of giant river prawns after challenge with *Aeromonas hydrophila*. The values are shown as means and SDs, n = 6. Data with different letters (a, b and c) were significantly different (*p* < 0.05) at the same exposure times. Descriptions of this part are also provided for Figure 9.

**Figure 9 biomolecules-11-01034-f009:**
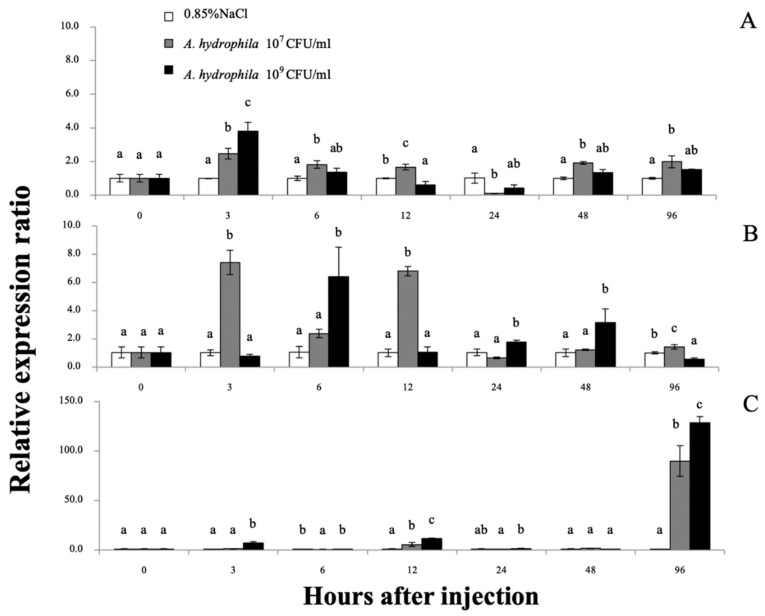
The *Mr-hsp90* mRNA expression levels relative to *β-actin* levels analyzed by qRT-PCR. in the gills (**A**), hepatopancreas (**B**), and hemocytes (**C**) of giant river prawns after challenge with *Aeromonas hydrophila*. Data with different letters (a, b and c) were significantly different (*p* < 0.05) at the same exposure times.

**Figure 10 biomolecules-11-01034-f010:**
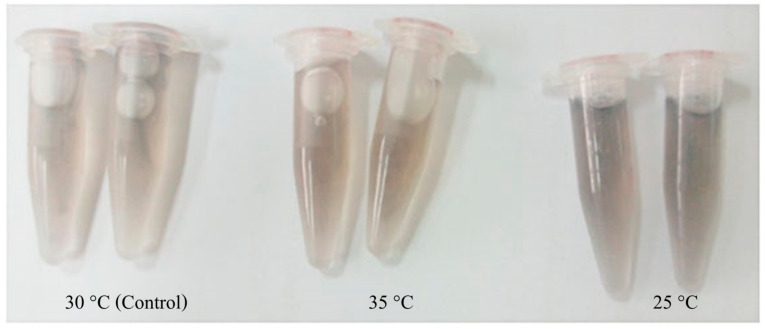
The colors of prawn hemolymph under different temperature inductions.

**Figure 11 biomolecules-11-01034-f011:**
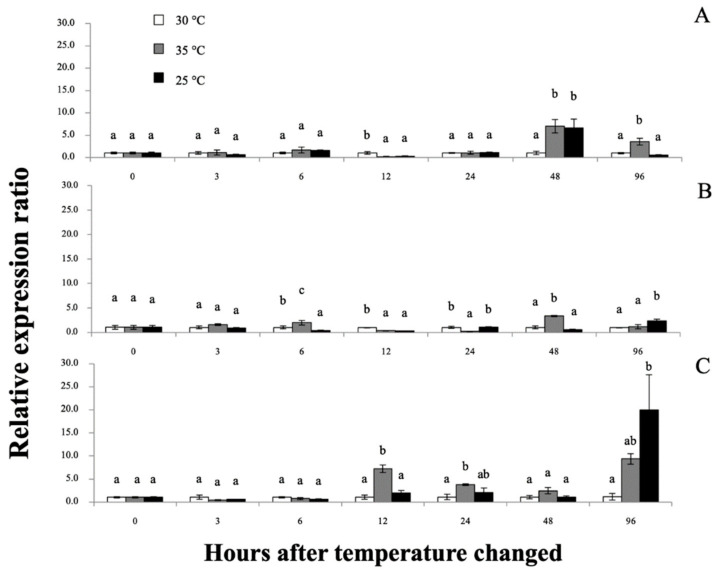
The *Mr-hsp40* mRNA expression levels relative to *β-actin* levels analyzed by qRT-PCR in the gills (**A**), hepatopancreas (**B**) and hemocytes (**C**) of giant river prawns after induction with temperatures at 30, 35 and 25 °C. The values are shown as means and SDs, n = 6. Data with different letters (a, b and c) were significantly different (*p* < 0.05) at the same exposure times. Descriptions of this part are also provided for Figure 12.

**Figure 12 biomolecules-11-01034-f012:**
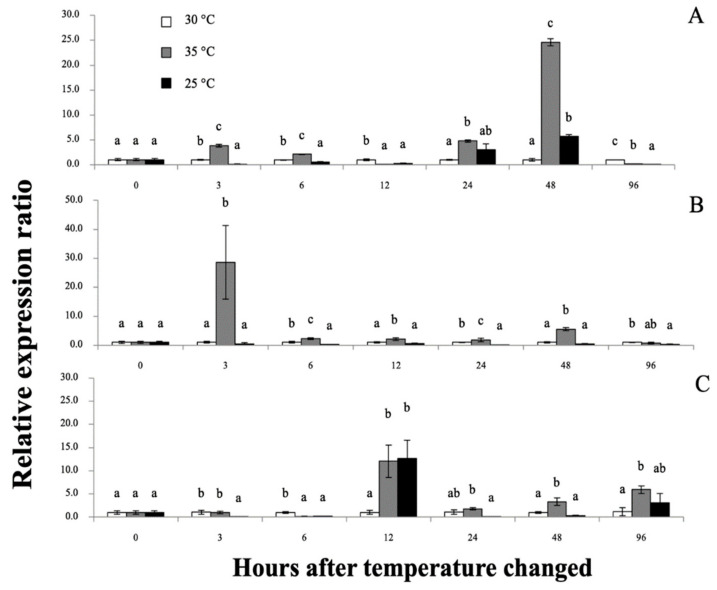
The *Mr-hsp90* mRNA expression levels relative to *β-actin* levels analyzed by qRT-PCR in the gills (**A**), hepatopancreas (**B**) and hemocytes (**C**) of giant river prawns after induction with temperatures at 30, 35 and 25 °C. Data with different letters (a, b and c) were significantly different (*p* < 0.05) at the same exposure times.

**Figure 13 biomolecules-11-01034-f013:**
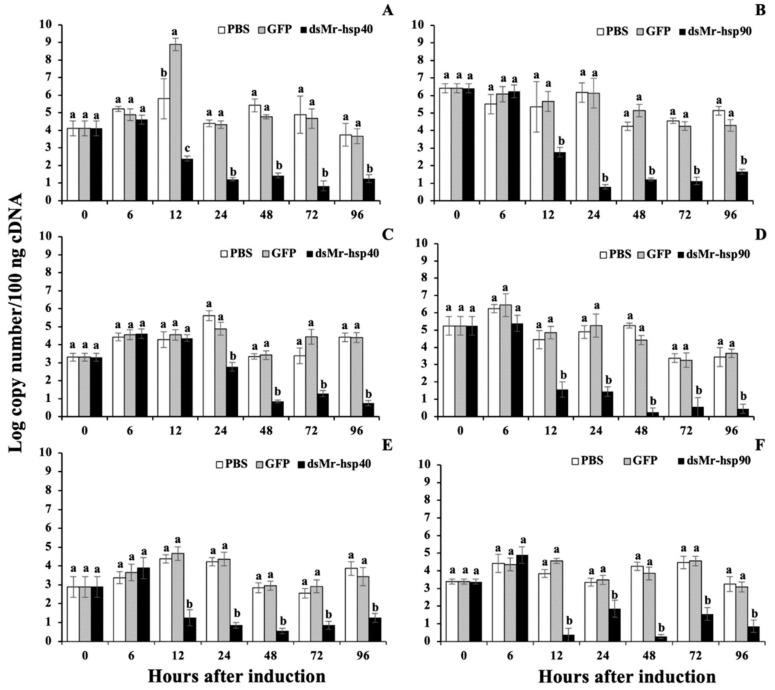
Silencing analysis of the *hsp40* and ds*Mr-hsp90* genes induced by ds*Mr-hsp40* and ds*Mr-hsp90* injection in the gills (**A**,**B**), hepatopancreas (**C**,**D**) and muscle (**E**,**F**). Data with different letters (a, b and c) were significantly different (*p* < 0.05) at the same time intervals.

**Figure 14 biomolecules-11-01034-f014:**
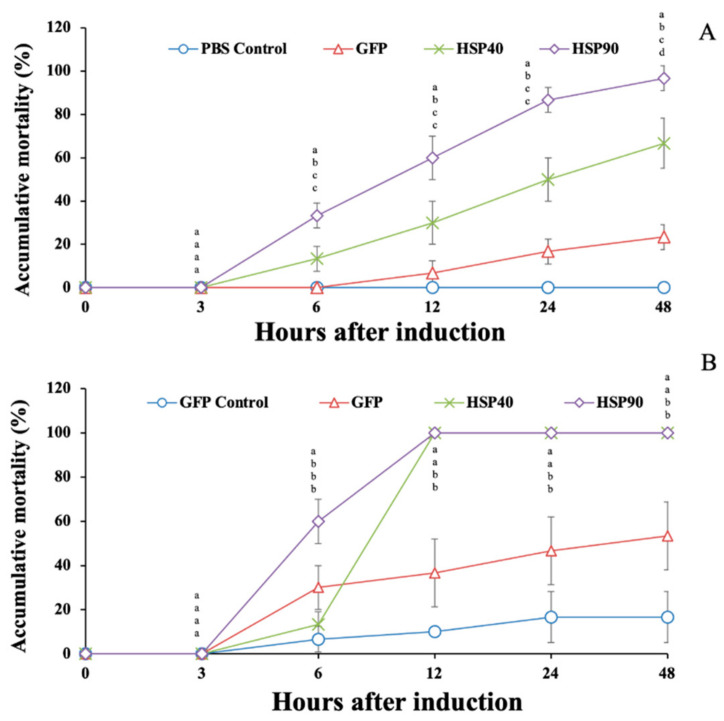
Effect of *hsp40* and ds*Mr-hsp90* gene knockdown on high-temperature (35 °C) (**A**) and *A. hydrophila* infection conditions (**B**). Data with different letters (a, b and c) were significantly different (*p* < 0.05) at the same time intervals.

**Table 1 biomolecules-11-01034-t001:** PCR primers and primer sequences used for the amplification of *hsp40*, *hsp90* and *β-actin* gene transcripts.

Gene	Primer Name	Primer Sequence (5’-3’)	Size	Purposes
*β-actin*	RT*B-actin*_F	TTCACCATCGGCATTGAGAGGTTC	119 bp	Real-Time PCR
	RT*B-actin*_R	CACGTCGCACTTCATGATGGAGTT		Real-Time PCR
*Mr-hsp40*	F_GFP_*hsp40*	TGCAGGTTCGCATACAACAGTTGG	398 bp	RT-RCR
	R_GFP_*hsp40*	TTCGCCTGGAATAGCACTGATCAC		RT-RCR, Touchdown PCR in RACE
	RTF_GFP_*hsp40*	TCGCAAGCATACGAAGTTCTCAGC	136 bp	Real-Time PCR
	RTR_GFP_*hsp40*	CCGAAGAACATGTCGAAGATGTCC		Real-Time PCR
	Mr-HSP40T7_F	GGATCCTAATACGACTCACTATAGGGGGTGGAAAGAAGCCCCCAG		Gene silencing
	Mr-HSP40T7_R	GGATCCTAATACGACTCACTATAGGGCTCCTGGTCTCCCTCACCA		Gene silencing
*Mr-hsp90*	F_GFP_*hsp90*	CGAGTACCTGAACTTCCTCAATGG	565 bp	RT-RCR
	R_GFP_*hsp90*	CCTGTTCACGAGATTCACCAGTGA		RT-RCR, Touchdown PCR in RACE
	RTF_GFP_*hsp90*	GATGAGATGTGCTCCCTGAAGGAT	91 bp	Real-Time PCR
	RTR_GFP_*hsp90*	CCTGTTCACGAGATTCACCAGTGA		Real-Time PCR
	Mr-HSP90T7_F	GGATCCTAATACGACTCACTATAGGTGAGGCCTGACCATGGGGAA		Gene silencing
	Mr-HSP90T7_R	GGATCCTAATACGACTCACTATAGGGGCGACAAGTCTGTCCAGCA		Gene silencing
*Green fluorescence protein* (*GFP*)	GFP_F	TAATACGACTCACTAAGGGAGACACATGAAGCAGCACGACCT		Gene silencing
	GFP_R	TAATACGACTCACTATAGGGAGAAGTTCACCTTGATGCCGTTC		Gene silencing
UPM	LongUPM primer	CTAATACGACTCACTATAGGGCAAGCAGTGGTAACAACGCAGAGT		Touchdown PCR in RACE
	ShortUPM primer	CTAATACGACTCACTATAGGGC

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
