# Peer review of "Functional and Stress Response Analysis of Heat Shock Proteins 40 and 90 of Giant River Prawn (Macrobrachium rosenbergii) under Temperature and Pathogenic Bacterial Exposure Stimuli"

_biomolecules, 2021, doi:10.3390/biom11071034_

Round 1
Reviewer 1 Report
The manuscript by Ju-Ngam and co-workers describes the analysis of HSP40 and Hsp90 in the Giant river Prawn Macrobrachium rosenbergii, De Man. The two chaperones were cloned and their expression under various conditions was characterized. The authors demonstrated a crucial role for the chaperones in both heat stress and bacteria invasion. Furthermore, RNAi experiments revealed an essential role for the chaperones under normal conditions as well, suggesting a role in the normal life cycle of the prawn. In general, these expression patterns were similar to those shown in previous studies for many organisms, from yeast to humans.
The cloning, phylogenesis, and expression methods are all standard methods in molecular biology and the data presentation is clear. Still, the data presentation could benefit from the use of in-graph significance analysis. This will highlight significant findings within the presented data.
Author Response
The manuscript by Ju-Ngam and co-workers describes the analysis of HSP40 and Hsp90 in the giant river prawn Macrobrachium rosenbergii, De Man. The two chaperones were cloned, and their expression under various conditions was characterized. The authors demonstrated a crucial role for the chaperones in both heat stress and bacteria invasion. Furthermore, RNAi experiments revealed an essential role for the chaperones under normal conditions as well, suggesting a role in the normal life cycle of the prawn. In general, these expression patterns were similar to those shown in previous studies for many organisms, from yeast to humans.
The cloning, phylogenesis, and expression methods are all standard methods in molecular biology and the data presentation is clear. Still, the data presentation could benefit from the use of in-graph significance analysis. This will highlight significant findings within the presented data.
Response: Thank you for your kind guidance on our research works. In the current version of our manuscript, we have carefully tried to address your concerns and provided more crucial details as the suggestion of the reviewer to improve the quality of our manuscript.
Reviewer 2 Report
Review for the Manuscript biomolecules-1275227
Title: Functional and Stress Response Analysis of Heat Shock Proteins 40 and 90 of Giant River Prawn (Macrobrachium rosenbergii, De Man) under Temperature and Pathogenic Bacterial Exposure Stimuli
General Comments: The manuscript describes the identification and characterization of two genes encoding heat-shock proteins (Hsp40 and Hsp90) in the giant river prawn Macrobrachium rosenbergii. Through the use of RT-qPCR, the authors explore the two genes distribution in various tissues and track their response to variations in temperature and pathogenic bacterial exposure to further characterize their biofunctional role. Together with gene knockdown results, the authors proposed that the new generated Mr-hsp40 and Mr-hsp90 here are crucially involved in cellular stress responses to both temperature shock and pathogenic bacterial stimuli. The topic certainly deserves much attention and the results obtained are is promising and are potentially of broad interest to relevant research community. The introduction, the materials and the method sections were generally well written and contained necessary information. Please consider the following comments when the manuscript is revised.
- The basis for selecting the two heat shock protein, Hsp40 and Hsp90, in the present study is unclear.
- Line 4: delete “De Ma”.
- Line 52: add reference(s) after “approximately 14-35°C”.
- The RACE experiment was performed in non-axenic conditions. Since Hsp40 and Hsp90 are highly conserved protein families, Hsp90 in particular, the authors should indicate that Mr-hsp40 and Mr-hsp90 yielded here are encoded by the DNA of Macrobrachium rosenbergii.
- Can authors please provide the touchdown PCR protocol? These can occasionally be more specific than the general PCR protocol (listed in Lines 128-140).
- The methods of “2.5. Phylogenetic analysis of hsp genes of various animal species” in Lines 167-173 are insufficient. Please provide more detailed information.
- Lines 134-135: The preparation of the template used in 5’-RACE is murky. Please clarify or revise.
- Did the authors find any difference in the morphology of the Macrobrachium rosenbergii due to stress (temperature and pathogenic bacterial exposure) ?
- The authors used β-actin as a reference gene for the quantitative PCR approach. Actin may not be a good choice for some organisms. It is not described whether the feasibility of actin as a reference gene in Macrobrachium rosenbergii was checked under the chosen conditions.
- The resolution of Figure 1, Figure 2, Figure 3, Figure 4 is low.
- Figure 5, Figure 6 seems incomplete and bootstrap values is written up to 50%
- I think that the expression: "Hsp40 and Hsp90” indicates the proteins. If so, the gene symbols should be in roman type.
- Discussion: I suggest some discussion on this aspect to tie the study back to the environment more coherently. These types of studies are certainly useful for contextualizing genetic response to changing environmental parameters, but I think it is important to also tie back to what the cell really experiences in the wild and how the study informs on the species in this context.
Author Response
General Comments: The manuscript describes the identification and characterization of two genes encoding heat-shock proteins (Hsp40 and Hsp90) in the giant river prawn Macrobrachium rosenbergii. Through the use of RT-qPCR, the authors explore the two genes distribution in various tissues and track their response to variations in temperature and pathogenic bacterial exposure to further characterize their biofunctional role. Together with gene knockdown results, the authors proposed that the newly generated Mr-hsp40 and Mr-hsp90 here are crucially involved in cellular stress responses to both temperature shock and pathogenic bacterial stimuli. The topic certainly deserves much attention and the results obtained are is promising and are potential of broad interest to the relevant research community. The introduction, the materials, and the method sections were generally well written and contained the necessary information. Please consider the following comments when the manuscript is revised.
- The basis for selecting the two heat shock proteins, Hsp40 and Hsp90, in the present study is unclear.
Response: Thank you so much. We have clarified this concern point in the last paragraph of “Introduction”.
- Line 4: delete “De Man”.
Response: “De Man” has been deleted.
- Line 52: add a reference(s) after “approximately 14-35°C”.
Response: The counterpart reference has been added; see reference [2].
- The RACE experiment was performed in non-axenic conditions. Since Hsp40 and Hsp90 are highly conserved protein families, Hsp90 in particular, the authors should indicate that Mr-hsp40 and Mr-hsp90 yielded here are encoded by the DNA of Macrobrachium rosenbergii.
Response: Thank you so much for the critical comments. Technically, we agree with this suggestion of the reviewer, and we always carefully consider this critical point. In our lab, we cloned and characterized various Mr-hsp40s and Mr-hsp90 of giant river prawn, and several Mr-hsp40s and Mr-hsp90 members have been identified. Therefore, to avoid the miss amplifying of the target gene fragments, all specifical primers have been carefully designed based on the non-conserved nucleotide regions of the target genes.
- Can the authors please provide the touchdown PCR protocol? These can occasionally be more specific than the general PCR protocol (listed in Lines 128-140).
Response: We have added this crucial information into section “2.3.5”.
- The methods of “2.5 Phylogenetic analysis of hsp genes of various animal species” in Lines 167-173 are insufficient. Please provide more detailed information.
Response: Thank you so much for your kind comments. Information in this part has been appropriately modified.
- Lines 134-135: The preparation of the template used in 5’-RACE is murky. Please clarify or revise.
Response: This part has been properly modified.
- Did the authors find any difference in the morphology of the Macrobrachium rosenbergii due to stress (temperature and pathogenic bacterial exposure)?
Response: Thank you so much for reminding us to put this important information. This suggestion has already been added into sections “3.4.1” and “3.4.2”.
- The authors used β-actin as a reference gene for the quantitative PCR approach. Actin may not be a good choice for some organisms. It is not described whether the feasibility of actin as a reference gene in Macrobrachium rosenbergii was checked under the chosen conditions.
Response: Thank you so much for this suggestion. Based on the report by Priyadarshi et al. (2015) and our previous research works. We confirm that β-actin is one of the most suitable candidates housekeeping gene for validation and normalization in RT- or qRT-PCR analysis in the giant river prawn compared to EF1α, GAPDH, and 18S rRNA. The low precision of normalization of β-actin was found when the tissues from the nerve cord were used, and therefore this tissue was not included in the expression responses of our present study.
- The resolution of Figure 1, Figure 2, Figure 3, Figure 4 is low.
Response: Thank you for this comment. We have enhanced the resolution of these Figures.
- Figure 5, Figure 6 seems incomplete and bootstrap values is written up to 50%
Response: Thank you for this suggestion. We agree well with this comment. When taking a look at Figure 5, the bootstrap was acceptable. However, Figure 6 showed a bit low bootstrap values at 49 on the major knot. We tried very hard to retest these results, but it still shows similar information as the previous tree. The low bootstrap values may cause by the high evolutionary distance between organism groups, especially from prokaryotes.
- I think that the expression: "Hsp40 and Hsp90” indicates the proteins. If so, the gene symbols should be in roman type.
Response: Thank you for this comment; we have corrected it throughout the manuscript.
- Discussion: I suggest some discussion on this aspect to tie the study back to the environment more coherently. These types of studies are certainly useful for contextualizing genetic response to changing environmental parameters, but I think it is important to also tie back to what the cell really experiences in the wild and how the study informs on the species in this context.
Response: Thank you for this insightful comment. Some crucial discussions have been properly revised in this part.
Round 2
Reviewer 1 Report
My recommendation for rejection is not based on poor scientific merit and therefore I cannot provide detailed revision to the authors. What I wrote that the findings in the manuscript are not significant enough to merit publication in Biomolecules since the importance of the Hsp40/70 chaperones under stress response was already demonstrated in many organisms from yeast to human. I, therefore, expect little interest from the readers
Reviewer 2 Report
No comment.